MFI-Net: multi-level feature invertible network image concealment technique

Cheng Dapeng 1 chengdapeng@sdtbu.edu.cn
http://orcid.org/0009-0000-3621-2142 Zhu Minghui 1
Yang Bo 2
Gao Xiaolian 1
Jing Wanting 3
Mao Yanyan 1
Zhao Feng 1
1 School of Computer Science and Technology, Shandong Technology and Business University , Yantai, Shandong , China
2 School of Information Technology, Hainan College of Economics and Business University , Haikou, Hainan , China
3 School of Information and Electronic Engineering, Shandong Technology and Business University , Yantai, Shandong , China
Coelho Paulo Jorge
Electronic publication date: 2025 Feb 14
Publication date: 2025
Volume: 11
Electronic Location ID: e2668
Received 2024 Aug 29; Accepted 2025 Jan 3
Copyright: © 2025 Cheng et al.
Copyright year: 2025
Copyright holder: Cheng et al.
License: This is an open access article distributed under the terms of the Creative Commons Attribution License, which permits unrestricted use, distribution, reproduction and adaptation in any medium and for any purpose provided that it is properly attributed. For attribution, the original author(s), title, publication source (PeerJ Computer Science) and either DOI or URL of the article must be cited.
License URL: https://creativecommons.org/licenses/by/4.0/

Keywords: Invertible network, Image steganography, Multi-level feature fusion, Frequency domain hiding, High quality, Strong generalization capability

Funding: National Natural Science Foundation of China 62176140 This work was supported by the National Natural Science Foundation of China (62176140). There was no additional external funding received for this study. The funders had no role in study design, data collection and analysis, decision to publish, or preparation of the manuscript.

==============================
The utilization of deep learning and invertible networks for image hiding has been proven effective and secure. These methods can conceal large amounts of information while maintaining high image quality and security. However, existing methods often lack precision in selecting the hidden regions and primarily rely on residual structures. They also fail to fully exploit low-level features, such as edges and textures. These issues lead to reduced quality in model generation results, a heightened risk of network overfitting, and diminished generalization capability. In this article, we propose a novel image hiding method based on invertible networks, called MFI-Net. The method introduces a new upsampling convolution block (UCB) and combines it with a residual dense block that employs the parametric rectified linear unit (PReLU) activation function, effectively utilizing multi-level information (low-level and high-level features) of the image. Additionally, a novel frequency domain loss (FDL) is introduced, which constrains the secret information to be hidden in regions of the cover image that are more suitable for concealing the data. Extensive experiments on the DIV2K, COCO, and ImageNet datasets demonstrate that MFI-Net consistently outperforms state-of-the-art methods, achieving superior image quality metrics. Furthermore, we apply the proposed method to digital collection images, achieving significant success.

Introduction

Steganography is a technique used to conceal information by embedding secret data within seemingly innocuous media, enabling the transmission of information without raising suspicion. Unlike encryption, the primary goal of steganography is not to protect the confidentiality of the message, but rather to hide its existence (Cheddad et al., 2010; Provos & Honeyman, 2003). Steganography can be applied to various types of media, including images, audio, video, etc. Image steganography, in particular, takes two images as input: one serves as the cover image, and the other as the secret image. The output is a stego image, which allows only informed recipients to recover the secret image, making it imperceptible to others. For security reasons, the stego image is typically required to be indistinguishable from the cover image (Jing et al., 2021). Image hiding methods play a significant role in various fields, including privacy protection, digital watermarking and covert communication, etc. Recently, with the rise of the digital collectibles industry, image hiding methods can be employed to embed copyright information, historical context, and other relevant details into digital collectibles.

Existing steganography techniques can be broadly categorized into traditional methods and deep learning-based methods. Traditional steganography methods typically can conceal only a small amount of information and are unable to meet the demands of hiding large amounts of data (such as several kilobytes or more) (Barni, Bartolini & Piva, 2001; Hsu & Wu, 1999). Representative methods include LSB steganography (Tamimi, Abdalla & Alallaf, 2013), which hides secret information by replacing the least significant bit of image pixels; statistical-based steganography, which embeds secret information by modifying the image’s color histogram or the statistical distribution of pixels (Qin et al., 2013; Tsai, Hu & Yeh, 2009); and file-based steganography, which embeds data altering the frequency spectrum of audio signals (Dutta et al., 2020; Gopalan, 2003), among others. With the rapid development of deep learning technology, steganography techniques based on deep learning have emerged. In 2017, Hayes & Danezis (2017) introduced adversarial training for steganography, competing with state-of-the-art methods. In 2018, Zhu et al. (2018) presented hiding data with deep networks (HiDDeN), a model robust to disturbances like Gaussian blur and JPEG compression. In 2020, Muhuri, Ashraf & Goel (2020) proposed a steganographic method based on integer wavelet transform for enhanced security and imperceptibility. In 2021, Tan et al. (2021) introduced an end-to-end GAN-based steganography network with a channel attention mechanism, achieving an embedding rate over four bits per pixel. These methods have significantly enhanced the quality and robustness of generated stego images. However, they generally suffer from limited capacity. To address this limitation, some researchers have proposed image steganography methods capable of hiding larger data. In 2017, Baluja (2017) was the first to apply convolutional neural networks (CNNs) for image hiding. Followed by Hayes & Danezis (2017), who proposed a GAN-based method. Jing et al. (2021) introduced Hi-Net, a framework based on invertible neural networks (INNs) that addresses capacity, invisibility, and security. Lu et al. (2021) proposed the Invertible Steganography Network (ISN) for large-capacity hiding. Yang et al. (2024) focused on robustness with the PRIS model, while Hu et al. (2024) used the Stable Diffusion model for robust generative image steganography. Yao et al. (2024) proposed an end-to-end GAN and Discrete Wavelet Transform (DWT)-based approach to improve the quality of the generated results. Su, Ni & Sun (2024) developed the Steganography StyleGAN for security, capacity, and quality.

Despite the outstanding performance of existing image hiding methods, there are still some challenges that need to be addressed. Most mainstream methods primarily rely on either pure deep convolutional structures or pure residual structures. Although these structures can enhance the network’s learning ability, they are prone to causing overfitting, which diminishes the network’s generalization ability. Additionally, excessive reliance on pure deep convolutional or residual structures may lead to insufficient attention to low-level image features, resulting in a decline in the quality of the generated results. Furthermore, selecting the most suitable regions in the cover image for embedding secret information is crucial. Appropriate hiding regions can make the secret information harder to detect by the human eye and improve the quality of the generated image. Most existing image hiding methods do not address this issue. The Hi-Net (Jing et al., 2021) method attempts to select more suitable regions for hiding secret information in the frequency domain after wavelet transformation of the cover image, but its selection remains insufficiently accurate.

To address these challenges, we propose an image hiding method called MFI-Net, inspired by existing techniques based on invertible neural networks. Our method aims to extract low-level image features and integrate them with high-level features to capture multi-level representations, thereby improving the quality of the network’s generated results. Additionally, the proposed method employs a novel loss function to identify and select the most suitable regions in the cover image for embedding secret information. Compared to hiding secret information in other regions, using the regions selected by this loss function effectively enhances the visual quality of the generated images and minimizes perceptible distortions during the hiding process. Extensive experiments demonstrate that MFI-Net not only achieves a large hiding capacity but also significantly enhances image quality and invisibility. Moreover, we have showcased a practical application of the proposed method, in which QR code images are hidden within digital collectible images to safeguard their copyright. The main contributions of this article are as follows: We propose a novel image hiding method, MFI-Net, which efficiently learns multi-level features of the image and more accurately selects the most suitable regions in the cover image for embedding secret information.

We have designed a novel upsampling convolution block (UCB) for the efficient extraction of low-level image features. A five-layer residual dense block containing the parametric rectified linear unit (PReLU) activation function is then used to extract high-level features, which are fused with the low-level features to significantly improve image quality.

We introduce a novel frequency domain loss (FDL) to more accurately identify the regions in the cover image that are most suitable for hiding secret information, thereby enhancing the quality and invisibility of the model’s generated results.

We demonstrate a practical application of the proposed method: copyright protection for digital collectible images.

In the following sections, this article will discuss the design and implementation of the proposed method. “Related Work” reviews the related work. “Method” details the proposed method and its loss function. “Experiments” presents the experimental parameter settings, dataset usage, and results. “Practical Applications of the Proposed Model” demonstrates a practical application of the proposed method. Finally, conclusions are presented in “Conclusion”.

Related work

Steganography is a technique used to hide one form of message, such as text or images, within another, without arousing suspicion. Image hiding technology, in particular, is the method of embedding one image within another and recovering it, a technique that has been extensively studied in the academic community (Chanu, Singh & Tuithung, 2012; Kadhim et al., 2019; Bender et al., 1996; Fridrich, Goljan & Du, 2001; Johnson & Jajodia, 1998; Chan & Cheng, 2004; Pevny & Fridrich, 2007; Kodovskỳ & Fridrich, 2009; Luo, Huang & Huang, 2010; Solanki, Sarkar & Manjunath, 2007). In this section, we offer a concise overview of traditional steganography, deep learning-based steganography, and image hiding technologies, along with relevant topics on invertible neural networks (INNs). Furthermore, we discuss the limitations of current image hiding techniques and explore potential solutions.

Traditional steganography

Traditional steganography includes a variety of methods, with some of the most representative approaches including least significant bit (LSB) (Tamimi, Abdalla & Alallaf, 2013), pixel value differencing (PVD) (Pan, Li & Yang, 2011; Wu & Tsai, 2003), histogram shifting (Qin et al., 2013; Tsai, Hu & Yeh, 2009), and Multi-bit Plane techniques (Nguyen, Yoon & Lee, 2006). Among these, the LSB algorithm utilizes the least significant bits of pixels or samples in images or audio to embed hidden information, enabling covert transmission. Pixel value differencing (PVD) hiding algorithm modifies the values of pixels in an image to conceal secret information. The Multi-bit Plane algorithm employs the color components of an image for hiding, embedding secret information into multiple bit planes of the image, providing high hiding capacity and concealment.

Deep learning-based steganography and image hiding techniques

Traditional steganography methods, such as LSB and PVD, are relatively simple to implement. However, they are limited in terms of hiding capacity, security, and result quality. In recent years, the rapid development of deep learning has led some researchers to propose high-performance, deep learning-based steganography methods (Zhang et al., 2023a, 2024, 2023b). Specifically, Hayes & Danezis (2017) introduced adversarial training for steganography, competing with state-of-the-art methods. Zhu et al. (2018) introduced HiDDeN, an end-to-end trainable data hiding framework that involves joint training of encoder and decoder networks. HiDDeN can be applied to both steganography and watermarking tasks. Muhuri, Ashraf & Goel (2020) proposed a new steganographic method based on the integer wavelet transform, offering strong security, imperceptibility, and robustness. In 2021, Tan et al. (2021) introduced channel attention image steganography with generative adversarial networks (CHAT-GAN), which incorporates an additional channel attention module to identify favorable channels in feature maps for information hiding. In 2024, Yu et al. (2023) proposed diffusion model makes controllable, robust and secure image steganography (CRoSS), a diffusion model-based framework for information hiding that achieved promising results.

Image steganography is an important branch of steganography, aiming to conceal an entire image within another. Baluja (2017) first used deep neural networks to hide a full-color image within another image. In 2021, Lu et al. (2021) introduced the ISN model, innovatively utilizing the invertible network method to hide single or multiple images in the pixel domain. In the same year, Jing et al. (2021) presented the Hi-Net model, a deep learning-based image hiding model that hides secret information in the frequency domain of the image. In 2024, Hu et al. (2024) proposed a novel and robust generative image steganography scheme. This method uses the popular Stable Diffusion model as the backbone generative network, enabling zero-shot text-driven stego image generation. Yao et al. (2024) proposed a end-to-end deep neural network for image steganography, based on generative adversarial networks (GANs) and Discrete Wavelet Transformation (DWT). This method significantly enhances the quality of the generated images. Su, Ni & Sun (2024) proposed a generic generative image steganography scheme called Steganography StyleGAN. It achieves the practical objectives of security, capacity, and quality within a unified framework. Yang et al. (2024) proposed PRIS, a highly robust image hiding technique based on invertible neural networks, which has achieved significant results in terms of both robustness and practicality.

To better illustrate the development process of steganography, this article presents in Table 1 a comparison of several key methods, highlighting their advantages (characteristics) over previous approaches, as well as some of their limitations.

Table 1 Characteristics and limitations of key steganography methods.

Method	Characteristics	Limitations	
LSB (Tamimi, Abdalla & Alallaf, 2013)	One of the earliest steganography methods, simply implemented.	Limited hiding capacity, poor quality of generated results.	
4 bit-LSB	Larger capacity compared to LSB, allows for image hiding.	Generated image quality is low, with visible distortion.	
Baluja (2017)	First used deep learning neural networks for information hiding, resulting in higher quality compared to traditional methods.	Uses a encoder-decoder structure, with a relatively simple network model and insufficient learning capability.	
Hi-Net (Jing et al., 2021)	Based on INNs, secret information is hidden in the frequency domain of the cover image, significantly improving the quality and security of the generated results.	The selection of secret information hiding regions is still not accurate enough, and there is a lack of attention to the low-level features of the image.	
PRIS (Yang et al., 2024)	It has stronger robustness than other image hiding methods, with results less affected by noise or compression.	Complex model, lengthy training, and lacks focus on low-level features of the image and suitable hiding regions.	

Invertible neural network

Invertible neural network (INN) was initially proposed by Dinh, Krueger & Bengio (2014). These networks feature an invertible structure, with the encoding and decoding processes sharing the same set of parameters. This design enables the network to accurately reconstruct the input from the output, ensuring lossless information transmission during both forward and backward propagation. In recent years, INN has found widespread application in various domains of deep learning. Applications include image super-resolution (Liu et al., 2023), image scaling (Xiao et al., 2020), real-time manufacturing control (Roach et al., 2023), remote sensing data (Tu et al., 2024), and data hiding and privacy preservation (Wang et al., 2024; Shang et al., 2023; Wang et al., 2022; Jia et al., 2023; Yang et al., 2023).

Issues that need to be addressed

Despite significant advancements in image hiding methods based on deep learning, some issues remain: Existing deep learning-based image hiding methods primarily rely on pure residual structures, which leads to insufficient attention to low-level image features, resulting in a decline in the quality of the generated outputs.

Existing image hiding methods fail to select the regions for hiding secret information in the cover image with sufficient precision, or they completely neglect to choose appropriate regions for hiding secret information during the hiding process. As a result, some secret information is hidden in regions of the cover image unsuitable for information hiding, leading to a decline in the quality of the generated output.

To address these issues, we propose a novel image hiding method based on invertible networks, called MFI-Net. This method incorporates an innovative upsampling convolution block (UCB) to extract low-level features of the image, which are then fused with high-level features extracted using a residual dense block with PReLU activation functions. This enables the method to capture multi-level features of the image. Additionally, the method employs a frequency domain loss (FDL) to accurately identify more suitable regions for hiding secret information in the wavelet-transformed cover image. Experimental results on multiple datasets show that the proposed method achieves exceptional performance.

Method

In this section, we present a detailed description of the architecture of MFI-Net. This model fully utilizes the multi-level information of the image, while constraining the secret information to be hidden in regions of the cover image that are more suitable for concealing the data, achieving high-capacity image concealment with superior quality.

Network structure

The model structure of MFI-Net is illustrated in Fig. 1, which consists of wavelet transformation blocks and several consecutive invertible blocks. Each invertible block involves three interactions. During the hiding process, the input comprises both the cover image and the secret image. The images first undergo wavelet transformation, resulting in high-frequency wavelet subbands and low-frequency wavelet subbands. These subbands then pass through the invertible neural network for processing (entering through the first invertible block and outputting through the n-th invertible block). The processed information is subsequently transformed into the stego image and lost information l through the inverse wavelet transformation block.

Figure 1 The framework of MFI-Net.

Similarly, during the recovery process, the input consists of the stego image and auxiliary information w, with the output being the recovered secret image secret-rev and the recovered cover image cover-rev. However, it is important to note that in the reverse recovery process, the flow of information is the opposite of the forward hiding process. Specifically, the information will enter through the n-th invertible block and be output by the first invertible block.

To more clearly illustrate the entire forward hiding process of the model, the following description is provided in Algorithm 1.

Algorithm 1 Algorithm for the forward hiding stage.

1: Input: xcover,xsecret	
2: Output: xstego,l	
3:  xcover1←DWT(xcover)	
4:  xsecret1←DWT(xsecret)	
5: for i=1 to n do	
6:   xcoveri+1←Invblocki(xcoveri)	
7:   xsecreti+1←Invblocki(xsecreti)	
8: end for	
9:  xstego←IWT(xcovern+1)	
10: l←IWT(xsecretn+1)	
11: return l,xstego	

Here, xcover and xsecret represent the input cover image and secret image, respectively. xstego and l denote the output stego image and lost information, respectively. xcoveri and xsecreti refer to the input to the i-th invertible block, DWT denotes to the wavelet transformation block, IWT refers to the inverse wavelet transformation block, Invblocki refers to the i-th invertible block, and n is the total number of invertible blocks in the network.

To more clearly illustrate the entire reverse recovery process of the model, the following description is provided in Algorithm 2.

Algorithm 2 Algorithm for the reverse recovery stage.

1: Input: xstego,w	
2: Output: xcover−rev, xsecret−rev	
3:  xstegon+1←DWT(xstego)	
4:  wn+1←DWT(w)	
5: for i=n downto 1 do	
6:   xstegoi←Invblocki(xstegoi+1)	
7:   wi←Invblocki(wi+1)	
8: end for	
9:  xcover−rev←IWT(xstego1)	
10:  xsecret−rev←IWT(w1)	
11: return xcover−rev, xsecret−rev	

Here, xstego and w represent the input stego image and auxiliary information, respectively. xcover−rev and xsecret−rev denote the recovered cover image and the recovered secret image, respectively. xstegoi+1 and wi+1 refer to the input to the i-th invertible block, DWT denotes the wavelet transformation block, IWT refers to the inverse wavelet transformation block, Invblocki represents the i-th invertible block, and n is the total number of invertible blocks in the network.

Auxiliary information w

If we aim to hide a secret image within a cover image while ensuring the stego image as similar as possible to the cover image, a portion of the original cover image’s information will inevitably be displaced. This occurs because the capacity of an image is finite. The displaced information is referred to as the lost information, denoted as l. In the recovery process, if one attempts to extract the restored image secret-rev only from the stego image without considering l, it becomes an ill-posed problem. Without the guidance of l, millions of secret images can be recovered from the same stego image. Therefore, this article introduces auxiliary information w as one of the inputs in the recovery process to assist in recovering the unique secret image from the stego image. w is sampled from a case-agnostic Gaussian distribution, which follows the same distribution as lost information l, i.e., w∼N(μ0,σ02). During the training process, the network adjust its parameters based on the feedback from the loss function, ensuring that each sample in the distribution can effectively recover the secret information. In the experiments conducted in this article, μ0 is set to 0 and σ02 is set to 1. (w is case-agnostic, this conclusion has been proven in Jing et al., 2021).

Upsampling convolution block

The residual structure is essential in the process of image hiding using invertible neural networks. The residual structure can effectively extract high-level image features such as abstract concept information, spatial relationship features, etc. Additionally, it helps significantly alleviate the performance degradation caused by network depth. However, while the residual structure focuses on the transmission and reconstruction of high-level semantic information, it often neglects the sparsity and subtle variations in low-level features. This can affect tasks such as image inpainting and restoration, where retaining certain low-level features is essential for combining them with high-level features during image reconstruction, thus maintaining higher fidelity and clarity of details. Moreover, a network structured solely with residual blocks is more prone to overfitting.

Therefore, we designed an upsampling convolution block (UCB) to extract low-level features from the image, enhancing the network’s performance. The overall structure of the UCB is shown in Fig. 2 and consists of two convolutional layers with normalization and LeakyReLU activation functions, as well as an upsampling block. The input feature maps are first convolved with a 3×3 kernel using a stride of 1, followed by normalization and LeakyReLU activation to produce activated feature maps. These activated feature maps are then downsampled with a 3×3 convolutional kernel and a stride of 2, preserving the same channel count. The output feature maps undergo normalization and LeakyReLU activation once more. Finally, the upsampling layer enlarges the feature maps by a factor of two, yielding the final output feature maps. For clarity on how this module processes information, its algorithmic description is provided in Algorithm 3.

Figure 2 The structure of the upsampling convolution block.

Algorithm 3 Algorithm for the upsampling convolution block.

1: Input: xucbin	
2: Output: xucbout	
3:  xco1←Conv3x3S1(xucbin)	
4:  xbn1←BatchNorm(xco1)	
5:  xlr1←LeakyReLU(xbn1)	
6:  xco2←Conv3x3S2(xlr1)	
7:  xbn2←BatchNorm(xco2)	
8:  xlr2←LeakyReLU(xbn2)	
9:  xucbout←Upsample(xlr2)	
10: return xucbout	

Here, xucbin denotes the input to the UCB, while xucbout denotes its output. Conv3×3S1 represents a convolution with a kernel size of 3×3 and a stride of 1, whereas Conv3×3S2 denotes a convolution with a kernel size of 3×3 and a stride of 2. BatchNorm stands for the batch normalization layer, LeakyReLU refers to the LeakyReLU activation function, and Upsample denotes the upsampling block.

Residual dense block

In this model, we employ a residual dense block with PReLU activation functions to process images. This structure not only extracts high-level features, maintaining excellent performance as the network deepens, but also captures multi-scale features from the images. Notably, we combine the PReLU function with the residual dense block. Unlike traditional rectified linear unit (ReLU), the PReLU function introduces learnable parameters, enhancing the network’s adaptability and robustness. This addition effectively mitigates gradient vanishing and improves the network’s expressiveness and training efficiency. As shown in Fig. 3, the residual dense block used in this article consists of five layers, with PReLU activation functions in the first four layers. The results of the final layer are output directly.

Figure 3 The structure of the residual dense block.

Why choose five layers? The number of layers in the residual dense block plays a crucial role in the model’s overall performance. To determine the optimal number of dense block layers for our model, we analyzed and referenced several studies (He et al., 2016a, 2016b; Li et al., 2019; He et al., 2017; Shin, 2022; Bejani & Ghatee, 2021; Zhu et al., 2022; Liu & Liu, 2022), on the relationship between network depth and performance. These studies suggest that a suitable depth of dense blocks can enhance the model’s learning capacity and extract deep features from the data. However, an excessive number of layers in dense blocks may reduce effectiveness, increase training time, and heighten the risk of overfitting. Therefore, we initially established that the depth of the dense blocks should be kept between four and seven layers. To finalize the layer count, we referenced recent advanced image hiding methods (Jing et al., 2021; Lu et al., 2021; Xu et al., 2022; Yang et al., 2024) that employ residual dense block structures. These studies consistently showed that a five-layer residual structure performs optimally for image hiding and restoration tasks, effectively extracting deep features without creating an overly complex, hard-to-train network prone to overfitting. Consequently, this article adopts a five-layer structure for the residual dense block.

Invertible block

As shown in Fig. 1, within the invertible block, three interactions occur between the cover image and the secret image after wavelet transformation. To describe more clearly how the invertible blocks process information during the forward hiding process, this article provides the corresponding formulas. Specifically, ⊙ represents element-wise matrix multiplication, k denotes the k-th block, K1 and K2 represent the residual dense block and the upsampling convolution block, cat(a,b,c) represents concatenating tensors a and b along dimension c, ε signifies a 3 × 3 convolution, and ϕ represents the sigmoid function. For the k-th concealing block in the forward hiding process, the inputs are xcoverk and xsecretk, and the outputs xcoverk+1 and xsecretk+1 are expressed as follows:

(1) xcoverk+1=xcoverk+ε(cat(K1(xsecretk),K2(xsecretk),1))

(2) xsecretk+1=xsecretk⊙exp(ϕ(K1(xcoverk+1)))+ε(cat(K1(xcoverk+1),K2(xcoverk+1),1)).

In the reverse recovery process, the flow of information is reversed compared to the forward hiding process. This means that the way in which invertible blocks process information in the reverse recovery process is the opposite of that in the forward hiding process. To clarify how the invertible blocks handle information during the reverse recovery process, this article provides the corresponding formulas. Specifically, ⊙, k, K1, K2, cat(a,b,c), ε, and ϕ retain the same meanings as in the hiding process. For the k-th recovery block in the reverse recovery process, the inputs are wk+1 and xstegok+1, and the outputs wk and xstegok are expressed as follows:

(3) wk=(wk+1−ε(cat(K1(xstegok+1),K2(xstegok+1),1)))⊙exp(−ϕ(K1(xstegok+1)))

(4) xstegok=xstegok+1−ε(cat(K1(wk),K2(wk),1)).

Loss function

The total loss function of this model is comprised of three components: hiding loss, recovery loss, and frequency domain loss. The hiding loss is used to ensure that the stego image is as similar as possible to the cover image. The recovery loss is used to ensure that the secret image is as similar as possible to the secret-rev image. The frequency domain loss is employed to constrain the secret information to regions of the cover image that are most suitable for hiding, thereby preserving the invisibility of the generated image and further enhancing its quality.

Hiding loss and recovery loss

As our model is based on invertible neural networks, it is essential to use reconstruction loss to ensure the integrity and reversibility of the model. The objective of reconstruction loss is to minimize the reconstruction error between the model’s output and input, enabling the model to accurately reconstruct the input data. In this article, we chose the mean absolute error (MAE) loss function due to its robustness and consistency in the value domain. The formula for the hiding loss is as follows:

(5) LOSSHi=∑i=1NF(xcoveri,xstegoi)

where LOSSHi represents the total hiding loss, N denotes the total number of training samples, and i denotes the i-th training sample. F is used to measure the discrepancy between the cover image xcoveri and the stego image xstegoi. In this context, F refers to the MAE loss function. For the recovery loss, we express it as follows:

(6) LOSSRev=∑i=1NF(xsecreti,xsecret−revi).

Here, LOSSRev represents the total recovery loss, N denotes the total number of training samples, and i signifies the i-th training sample, F still denotes the MAE loss, which is used to measure the discrepancy between the genuine secret image xsecreti and the recovered secret image xsecret−revi.

Frequency domain loss

In the image hiding process, selecting the appropriate region for embedding the secret information is crucial. After performing the wavelet transform on the cover image, four subdomains are generated: High-High (HH), High-Low (HL), Low-High (LH), and Low-Low (LL). Among these, the HH domain contains the highest amount of high-frequency information, followed by LH and HL domains, and the LL domain contains the least. The more high-frequency information a subdomain contains, the more detailed and textured the information is. Since the human eye is less sensitive to high-frequency details and textures compared to low-frequency information, secret data hidden in high-frequency regions is less likely to be detected by the human eye than in low-frequency domains. Therefore, subdomains with higher high-frequency information are more suitable for embedding secret information than those with lower high-frequency content. Existing image hiding methods often lack precision in selecting the appropriate regions for hiding secret data, which can result in a deterioration of the quality of the generated images.

To address this issue, this article introduces a frequency domain loss (FDL) to ensure that secret information is prioritized for hiding in subdomains containing more high-frequency information. Specifically, FDL aims to constrain the LL domain of the cover image to be as identical as possible to the LL domain of the stego image, while maintaining a certain level of similarity between the LH and HL domains of both images (with the constraint strength being weaker than that of the LL domain). In this way, secret information is primarily hidden in the HH domain, as this domain is not constrained by the loss function. If the HH domain’s capacity is insufficient to store all the secret information, the excess will be hidden in the LH and HL domains, which are subject to weaker constraints. Only when the capacities of the HH, HL, and LH domains are all insufficient will the remaining secret information be hidden in the LL domain. The formula for the FDL is as follows:

(7) LOSSFD−LL=∑i=1NF(τ(xcoveri)LL,τ(xstegoi)LL)

(8) LOSSFD−LH=∑i=1NF(μ(xcoveri)LH,μ(xstegoi)LH)

(9) LOSSFD−HL=∑i=1NF(σ(xcoveri)HL,σ(xstegoi)HL)

(10) LOSSF=0.5∗(LOSSFD−HL+LOSSFD−LH)+LOSSFD−LL

where LOSSFD−LL represents the LL loss, LOSSFD−LH signifies the LH loss, LOSSFD−HL denotes the HL loss and LOSSF represents the total FDL. τ represents the operation of extracting LL frequency sub-bands after wavelet transformation, μ represents the operation of extracting LH frequency sub-bands after wavelet transformation, σ represents the operation of extracting HL frequency sub-bands after wavelet transformation, and F still denotes the MAE loss function.

Why is the weight for LOSSFD−HL+LOSSFD−LH set to 0.5? As mentioned above, FDL aims to constrain the LL, HL, and LH domains of the wavelet transform of both the cover and stego images, with the constraint on the LL domain being significantly greater than that on the HL and LH domains. Therefore, the weight of LOSSFD−HL+LOSSFD−LH should be less than 0.8, meaning the constraint on these domains should be significantly weaker than on LOSSFD−LL (where a weight of 1 implies equal constraints on the HH, HL, and LH domains). Otherwise, if the HH domain has insufficient capacity, the overflowed secret information will not be preferentially hidden in the LH and HL domains, leading to a deterioration in the quality of the generated results. In addition, the weight of LOSSFD−HL+LOSSFD−LH should not be less than 0.3, as this would result in an excessively weak constraint from the loss function, causing some secret information to be hidden in the HL or LH domains even when the HH domain still has sufficient capacity. This would, in turn, lead to a deterioration in the quality of the generated results. In conclusion, we can initially set the weight range for LOSSFD−HL+LOSSFD−LH to be between 0.3 and 0.7. To determine the final weight, we conducted experiments on 400 natural images from the COCO (2017 Test images) (Lin et al., 2014) dataset, testing weights between 0.3 and 0.7. The results are shown in Table 2, and a weight of 0.5 was found to be the optimal choice.

Table 2 The impact of different weight values on the results.

The bolded parts of the data represent the optimal parameters and results.

Values	Cover/Stego image pair	Secret/Secret-rev image pair	
PSNR (dB)	SSIM	PSNR (dB)	SSIM	
0.3	39.92	0.980	38.32	0.966	
0.4	40.12	0.983	38.06	0.958	
0.5	40.35	0.989	39.61	0.987	
0.6	39.94	0.981	38.12	0.962	
0.7	38.88	0.965	37.03	0.949	

Total loss

The total loss is the linear sum of the hiding loss, recovery loss, and frequency domain loss, where γ1, γ2, and γ3 represent their respective weights. The overall formula is as follows:

(11) LOSSTotal=γ1∗LOSSHi+γ2∗LOSSRev+γ3∗LOSSF.

During the training process, we use the gradient descent method to minimize the total loss, thereby forcing the model to achieve the desired outcomes.

Experiments

In this section, we provide an overview of the experimental setup and present the results used to evaluate the performance of the proposed method. This includes the datasets used for training and testing, the experimental configurations and parameters, the evaluation metrics, as well as both quantitative and qualitative results. Furthermore, we conduct statistical analysis and ablation studies to provide a comprehensive evaluation of the method and to investigate the contribution of various components of the proposed approach.

Experimental details

Experimental datasets

This article utilizes the DIV2K (https://data.vision.ee.ethz.ch/cvl/DIV2K/) (Agustsson & Timofte, 2017) training dataset for network training, which comprises 800 natural images. The testing dataset consists of 100 natural images from the DIV2K testing dataset, 400 natural images from the COCO (https://cocodataset.org/#download) (Lin et al., 2014) testing dataset, and 5,500 natural images from the ImageNet (https://www.image-net.org/challenges/LSVRC/index.php) (Russakovsky et al., 2015) testing dataset. All experiments were conducted on an NVIDIA GeForce RTX 3080 GPU with 10 GB of VRAM.

Data preprocessing

Due to the nature of the datasets used in this study (DIV2K, COCO, and ImageNet), which provide raw image data, preprocessing is necessary before model training and testing. Specifically, for the training dataset, all images are normalized to ensure that pixel values are within the range of 0 to 1, thereby enhancing training stability and model performance. Additionally, this study applies data augmentation techniques such as random cropping, horizontal flipping, and rotation to the training data. This approach increases data diversity and enhances the model’s generalization capability. For the testing dataset, we perform center cropping and standardization but do not apply data augmentation techniques like horizontal flipping and rotation. This ensures that the model’s performance in real-world conditions accurately reflects its capabilities.

Experimental parameters

This article resizes images to 224×224 for network training, uses a batch size of 4, sets the initial learning rate to 1∗10−3.9, reduces the learning rate to 1∗10−4.4 after five epochs, and then halves it every 260 epochs, with a total of 2,000 epochs. The model is trained using the Adam optimizer (Qin et al., 2013). For other model hyperparameters, the number of invertible blocks M is set to 18, the initial values of γ1, γ2, γ3 are all set to 1, with manual fine-tuned during multiple training iterations.

Evaluation metrics

This article selects Structural Similarity Index (SSIM), Peak Signal-to-Noise Ratio (PSNR), average percentage difference (APD), and root mean square error (RMSE) as the primary evaluation metrics. Compared to other metrics, these are more appropriate for image hiding tasks, as they assess image quality from various perspectives, offering a comprehensive view of the effects of image hiding and recovery.

Specifically, SSIM provides a quality assessment that aligns more closely with human visual perception by considering image structure. PSNR, as a widely used and easily computable standard, can quickly reflect the effectiveness of image reconstruction. APD focuses on revealing the relative error between predicted and actual values, effectively reflecting the model’s performance in detail handling. RMSE quantifies the overall error, offering an intuitive quantitative basis for model evaluation. The following is a detailed introduction to these metrics:

PSNR is a widely used metric for measuring image or video quality. It is employed to evaluate the similarity between the original signal, typically an image or video, and the reconstructed signal after compression, distortion, or other processing. The calculation formula is as follows, where MSE represents the mean squared error:

(12) PSNR=20log10(maxx)−10log10(MSE)

The SSIM is based on principles of human perception. It calculates similarity by comparing various aspects such as structure, luminance, and contrast between the original and reconstructed signals. SSIM takes into account not only the mean squared error but also losses in structural information and perceptual distortions. The formula is represented as follows:

(13) SSIM=(2μxμy+C1)(2σxy+C2)(μx2+μy2+C1)(σx2+σy2+C2).

APD represents the average percentage difference between predicted and actual values, reflecting the relative error in the predicted values compared to the actual values. The calculation formula is as follows:

(14) APD=1n∗∑i=1N|yi−y^iyi|∗100.

RMSE is based on the units of actual values and is used to measure the overall difference between predicted and actual values. The calculation formula is as follows:

(15) RMSE=1n∗∑i=1N(yi−y^i)2.

For PSNR and SSIM, a higher value indicates better results, whereas for APD and RMSE, a lower value signifies better performance.

Comparative experiments

Quantitative results

Table 3 compares our results with the methods of four bit-LSB (Baluja, 2017; Weng et al., 2019), practical robust invertible network for image steganography (PRIS) (Yang et al., 2024), joint adjustment image steganography networks (JAIS-Nets) (Zhang et al., 2023b), and HiNet (Jing et al., 2021). As shown in Table 3, it is evident that MFI-Net outperforms all other methods across all four metrics of cover/stego and secret/secret-rev image pairs. In the DIV2K dataset, the PSNR and SSIM of the cover/stego image pair generated by MFI-Net are 3.22 dB and 0.001 higher, respectively, than the second best results. The APD and RMSE are 0.96 and 1.22 lower, respectively, than the second-best results. For secret/secret-rev image pairs, MFI-Net also performs exceptionally well. Additionally, on the COCO (Lin et al., 2014) and ImageNet (Russakovsky et al., 2015) datasets, MFI-Net performs exceptionally well, demonstrating its strong generalization capability.

Table 3 Benchmark comparisons on different datasets.

The bolded parts of the data represent the best results.

Methods	DIV2K	
Cover/Stego image pair	Secret/Secret-rev image pair	
PSNR (dB)	SSIM	APD	RMSE	PSNR (dB)	SSIM	APD	RMSE	
4 bit-LSB	31.92	0.899	5.29	6.49	29.37	0.891	7.32	8.68	
Baluja (2017)	36.52	0.941	2.95	3.93	32.12	0.932	5.06	6.54	
Weng et al. (2019)	33.71	0.942	4.48	5.57	36.38	0.963	3.28	4.23	
PRIS (Yang et al., 2024)	39.17	0.961	2.18	2.90	41.01	0.988	1.74	2.4	
JAIS-Nets (Zhang et al., 2023b)	32.64	0.967	6.92	8.30	34.22	0.960	5.53	7.17	
Hi-Net (Jing et al., 2021)	48.99	0.997	1.33	1.94	52.86	0.999	0.56	0.86	
MFI-Net	52.21	0.998	0.37	0.72	53.55	0.999	0.25	0.58	
Methods	COCO	
Cover/Stego image pair	Secret/Secret-rev image pair	
PSNR (dB)	SSIM	APD	RMSE	PSNR (dB)	SSIM	APD	RMSE	
4 bit-LSB	31.88	0.886	5.36	6.53	29.58	0.919	7.10	8.49	
Baluja (2017)	35.47	0.928	3.32	4.45	32.46	0.949	4.99	6.27	
Weng et al. (2019)	30.85	0.917	6.80	8.33	35.58	0.965	3.53	4.67	
PRIS (Yang et al., 2024)	38.95	0.960	2.16	2.98	38.91	0.981	2.20	3.09	
JAIS-Nets (Zhang et al., 2023b)	28.79	0.943	11.19	13.06	30.81	0.930	7.09	9.67	
Hi-Net (Jing et al., 2021)	42.67	0.985	1.34	2.03	48.22	0.993	0.77	1.27	
MFI-Net	47.27	0.995	0.74	1.30	56.43	0.997	0.30	0.67	
Methods	ImageNet	
Cover/Stego image pair	Secret/Secret-rev image pair	
PSNR (dB)	SSIM	APD	RMSE	PSNR (dB)	SSIM	APD	RMSE	
4 bit-LSB	33.52	0.944	4.48	5.44	10.17	0.083	62.30	80.22	
Baluja (2017)	39.10	0.978	2.16	3.00	32.35	0.951	4.92	6.94	
Weng et al. (2019)	33.56	0.951	6.06	7.53	35.84	0.950	4.89	6.39	
PRIS (Yang et al., 2024)	38.07	0.965	2.31	3.35	36.96	0.976	2.59	3.86	
JAIS-Nets (Zhang et al., 2023b)	28.09	0.948	11.61	13.79	31.2	0.929	6.99	9.80	
Hi-Net (Jing et al., 2021)	38.66	0.984	1.91	3.20	38.58	0.983	1.94	3.22	
MFI-Net	40.35	0.989	1.47	2.66	39.61	0.987	1.62	2.91	

Statistical analysis

Box plot results. Figure 4 presents the box plots of PSNR results generated by MFI-Net and other different methods on the COCO dataset. The experimental results demonstrate that MFI-Net outperforms all other methods for both image pairs (Cover/Stego and Secret/Secret-rev). Specifically, the PSNR values of MFI-Net’s generated results are the highest and relatively concentrated, significantly outperforming other methods. In contrast, PRIS (Yang et al., 2024) and Hi-Net (Jing et al., 2021) also perform well, with relatively high and concentrated PSNR values. Other methods, however, have lower PSNR values or a wider distribution range, indicating relatively unstable performance.

Figure 4 Box plots of the generated results by different methods.

Analysis of variance. In addition to using box plots to display the distribution of PSNR across different methods, this article further quantifies the differences in steganographic performance between MFI-Net and other methods through analysis of variance (ANOVA), as shown in Fig. 5. It can be observed that, for both cover/stego image pairs and secret/secret-rev image pairs, the −log10(P-value) values between MFI-Net and the various methods are all significantly greater than 1.3 (the −log10 value at p = 0.05). This indicates that the improvement in steganographic performance of MFI-Net is significant and not due to chance, demonstrating MFI-Net’s exceptional performance and strong generalization capability in steganographic tasks.

Figure 5 Results of the analysis of variance for different methods.

Qualitative results

Figure 6 presents the quantitative results of each method. The images generated by the four bit-LSB, Baluja (2017), and Weng et al. (2019) methods exhibit varying degrees of visible distortion. Specifically, the four bit-LSB method produces images with significant distortion. Baluja’s (2017) method shows some improvement in quality, but noticeable blurring and artifacts remain. Weng et al.’s (2019) method generates higher-quality images, though slight distortions are still visible. In contrast, the images produced by JAIS-Nets (Zhang et al., 2023b), PRIS (Yang et al., 2024), Hi-Net (Jing et al., 2021), and MFI-Net exhibit almost no visible distortion. To illustrate this more clearly, the corresponding residual images are provided. From these residual images, it is evident that the images generated by MFI-Net are closest to the original images, with the least residual noise, demonstrating the best performance.

Figure 6 Comparison between stego images and secret-rev images generated by different methods, where the first column is the original images, starting from the second column, odd rows show the results generated by different methods, and even rows show the residual images between the generated images and the original images.

Ablation experiment

Effectiveness of the upsampling convolution block

In Table 4, we present the results of ablating the proposed UCB. It can be observed that, after removing the UCB, the PSNR and SSIM of the cover/stego image pair decreased by 3.68 dB and 0.007, respectively, while the APD and RMSE increased by 0.46 and 0.5. These changes highlight the effectiveness of the UCB module.

Table 4 The effectiveness of the UCB and FDL in MFI-Net.

The bolded parts of the data represent the best results.

UCB	FDL	Cover/Stego image pair	Secret/Secret-rev image pair	
PSNR (dB)	SSIM	APD	RMSE	PSNR (dB)	SSIM	APD	RMSE	
Yes	Yes	47.27	0.995	0.74	1.30	56.43	0.997	0.303	0.671	
No	No	42.38	0.979	1.88	2.72	48.90	0.994	0.693	1.192	
Yes	No	45.85	0.993	0.78	1.53	47.71	0.993	0.791	1.340	
No	Yes	43.59	0.988	1.20	1.85	56.39	0.997	0.261	0.647	

For the secret/secret-rev image pair, after removing this module, we observed a decrease of 0.04 dB in PSNR, no change in SSIM, but a decrease of 0.042 and 0.024 in APD and RMSE, respectively. Our investigation suggests that this is a normal occurrence, given the specific nature of invertible neural networks, where the quality of stego images and secret-rev images is relative. A better-hidden image implies greater difficulty in recovery. Most importantly, the improvement in the quality of stego images due to this module far outweighs the subtle effects on secret image recovery.

Effectiveness of frequency domain loss

Table 4 shows the results after removing the FDL. For the cover/setgo image pair, the PSNR and SSIM decreased by 1.42 dB and 0.002, while APD and RMSE increased by 0.04 and 0.23. For the secret/secret-rev image pair, the PSNR and SSIM decreased by 8.72 dB and 0.004, APD and RMSE increased by 0.488 and 0.669. These changes highlight the effectiveness of the proposed FDL in this study.

In addition to separately ablating the UCB and FDL, this article also tested the model with both components ablated. The results show that, compared to the ablated model, the original model generates a cover/stego image pair with PSNR and SSIM values higher by 4.89 dB and 0.016, respectively, while APD and RMSE are lower by 1.14 and 1.42. The generated secret/secret-rev image pair exhibited a higher PSNR and SSIM by 7.53 dB and 0.003, respectively, while APD and RMSE were reduced by 0.39 and 0.521. These results validate the effectiveness and compatibility of the UCB and FDL proposed in this article.

Figure 7 illustrates the results generated by the model after removing different modules along with their corresponding residual maps. It can be observed that for stego images, removing both the UCB and the FDL results in a decrease in image quality, with the UCB having a greater impact. For the recovered secret images, removing the FDL leads to a substantial reduction in image quality, while removing the UCB slightly enhances image quality. However, the improvement in stego image quality due to the UCB outweighs the minimal effect on secret image recovery. Therefore, the UCB is retained in the model.

Figure 7 Comparison between stego images and secret-rev images generated by MFI-Net after removing different modules, where the first column is the original image, starting from the second column, odd rows display the results generated by MFI-Net after removing different modules, and even rows display the residual images between the generated images and the original images.

Practical applications of the proposed model

With the rapid advancement of the internet and information science, steganography has gained widespread use in information protection and copyright management, particularly in applications such as QR code embedding. This technology aims to invisibly embed information into images, ensuring that important data can be transmitted without compromising the visual experience. However, current steganographic techniques face some challenges in practical applications. Some methods can generate high-quality steganographic images, but their data capacity is relatively limited, allowing only a small amount of text information to be hidden. This restriction fails to meet the demand for embedding large amounts of information (such as QR codes or other images). On the other hand, some methods may offer larger data capacities. However, the quality of the generated steganographic images or the recovered secret images is still not sufficiently high, resulting in a decrease in the readability of the hidden information.

In this context, MFI-Net provides an effective solution. According to experiments and research, the higher the SSIM value of an image (with a maximum of 1 indicating that two images are identical), the more difficult it is for the human eye to perceive differences between the two images. As shown in Table 3, MFI-Net achieves SSIM values of around 0.99 across various datasets, demonstrating its capability to conceal a secret image within another color image and recover it successfully. This ensures that the differences between the stego image and the recovered secret image are virtually imperceptible to the human eye. This feature enables the embedding of a QR code containing copyright information into a digital collection image without compromising the user’s visual experience (the digital collection image was purchased by the author on the New Bee collection platform).

The actual test results are shown in Fig. 8. By enlarging the images, it is evident that the quality of the hiding and recovery results generated by MFI-Net is superior. Compared to the other two methods, MFI-Net produced the best results with no visible distortion. In contrast, Baluja’s (2017) method exhibited distortion in both the stego image and the secret-rev image, while Hi-Net (Jing et al., 2021) generated a high-quality stego image but displayed slight distortion in the secret-rev image.

Figure 8 Results of embedding a QR code image into a digital collectible image using different methods.

Conclusion

This article proposes a novel image hiding method based on invertible neural networks, called MFI-Net. The method incorporates an upsampling convolution block to extract low-level features from images and combines it with a residual dense block that employs the PReLU activation function to capture multi-level features, addressing the limitation of existing methods that overlook low-level image details. Additionally, we introduce a novel frequency domain loss to constrain the secret information within the regions of the cover image that contain more high-frequency information after wavelet transformation. This enables more precise selection of suitable hiding regions. Compared to state-of-the-art models, MFI-Net generates higher-quality images while demonstrating superior generalization capabilities and practicality.

In the future, we plan to integrate generative adversarial networks with our approach, utilizing invertible neural networks as generators to further enhance the robustness of the method and reduce the detectability of the generated images. Another promising direction for exploration is the various regions used for hiding secret information in image hiding. While concealing images in the frequency domain is effective, it may not always be the optimal choice. We aim to investigate alternative spaces, such as color space or non-spatial domains, that can be integrated with invertible neural networks. Additionally, we plan to extend the method’s applicability to other fields, including video steganography, audio steganography, and document watermarking.

Supplemental Information

Supplemental Information 1 The code implementation of the paper.

Supplemental Information 2 Digital artwork image.

The digital artwork image was purchased and used for practical application testing.

Supplemental Information 3 The QR code image used for practical application testing.

The image was used for practical application testing, generated by a Python library.

We would like to thank the anonymous reviewers, the editor, and the staff, whose comments, suggestions, and support helped improve this manuscript.

Additional Information and Declarations

Competing Interests

The authors declare that they have no competing interests.

Author Contributions

Dapeng Cheng conceived and designed the experiments, authored or reviewed drafts of the article, and approved the final draft.

Minghui Zhu conceived and designed the experiments, performed the experiments, analyzed the data, performed the computation work, prepared figures and/or tables, authored or reviewed drafts of the article, and approved the final draft.

Bo Yang analyzed the data, authored or reviewed drafts of the article, and approved the final draft.

Xiaolian Gao analyzed the data, authored or reviewed drafts of the article, and approved the final draft.

Wanting Jing analyzed the data, authored or reviewed drafts of the article, and approved the final draft.

Yanyan Mao analyzed the data, authored or reviewed drafts of the article, and approved the final draft.

Feng Zhao analyzed the data, authored or reviewed drafts of the article, and approved the final draft.

Data Availability

The following information was supplied regarding data availability:

The raw data includes images from the public datasets:

- DIV2K (High Resolution Images), https://data.vision.ee.ethz.ch/cvl/DIV2K.

- COCO (2017 Test images), https://cocodataset.org/#download.

- ImageNet (ILSVRC 2017), https://www.image-net.org/challenges/LSVRC/index.php.

The complete code is available in the Supplemental File.

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
