# Peer review of "MFI-Net: multi-level feature invertible network image concealment technique"

_PeerJ Computer Science, doi:10.7717/peerj-cs.2668_

## Round 0.1 · original submission · Major Revisions

Dear authors,

You are advised to critically respond to all comments point by point when preparing an updated version of the manuscript and while preparing for the rebuttal letter. Please address all comments/suggestions provided by reviewers, considering that these should be added to the new version of the manuscript.

Kind regards,

**Language Note:** The review process has identified that the English language must be improved. PeerJ can provide language editing services - please contact us at [email protected] for pricing (be sure to provide your manuscript number and title). Alternatively, you should make your own arrangements to improve the language quality and provide details in your response letter. – PeerJ Staff

PCoelho

Reviewer 1 ·

Basic reporting

The work reported in this paper has several limitations. Kindly find the following issue.
1. The presentation of the work is not clear. A thorough proofreading is recommended.
2. The introduction is too much limited. It should provide a broader perspective of the field.
3. What is the contribution of the current study? Indicate in the introduction as bullet points.
4. I would suggest expanding the study and including recent papers published as indicated. This would provide a more clear trend.
5. Enhance the readability of the paper, in particular, transitions from section to section should be smoother.

Experimental design

As above

Validity of the findings

As above

Cite this review as

Reviewer 2 ·

Basic reporting

The manuscript is well-structured and covers relevant literature but would benefit from clearer articulation of the research gap and improved English language quality in certain sections for better readability. Figures and tables are relevant and informative but could be enhanced for clarity.

Experimental design

The experimental design is sound, with well-defined research objectives and use of appropriate datasets (DIV2K, COCO, ImageNet) to validate the model's performance. However, additional details on data preprocessing and rationale for chosen hyperparameters would improve reproducibility.

Validity of the findings

The findings are compelling, with MFI-Net showing significant improvements over existing methods across multiple metrics (PSNR, SSIM). Adding statistical significance tests and addressing potential limitations would further strengthen the results' validity and generalizability.

Additional comments

1. In section 1, Introduction section, Clarity of research problem and the novelty of MFI-Net over other invertible neural networks are missing.

2. In Section 2.2, Related Work, The deep learning-based steganography methods described are useful but could benefit from a table summarizing key methodologies, results, and limitations, to give readers a quick overview of advancements in this field.

3. In section 3.1, quality Figure 1 must be improved. Distinct labels for the forward hiding and reverse revealing processes are missing and also color-coding the layers could improve visualization.

4. In Section 3.1.2, Upsampling Convolution Block, description of the block could benefit from a pseudo-code outline or flow diagram (Figure 2) for better understanding.

5. In Section 3.1.3, Residual Dense Block, Clarify why five layers are chosen for this block. Referencing recent studies on optimal dense block configurations could strengthen this design choice.

6. In Section 3.2.2, Frequency Domain Loss, In Equation (10), explain the rationale behind choosing 0.57 as a coefficient. Justify if it is empirically derived or theoretically grounded, as it significantly impacts the final output.

7. In Section 4.1.3, Evaluation Metrics, discussing why certain metrics like SSIM and PSNR are prioritized over others. Adding a brief rationale for selecting each metric would strengthen the methodology.

8. In section 4.1, experimental Details, preprocessing steps for the datasets (DIV2K, COCO, and ImageNet) are missing. Describe image normalization or any data augmentation techniques used.

9. In Section 4.2.2, Statistical Analysis, The PSNR box plots in Figure 4 should include a statistical test (e.g., ANOVA) to verify the significance of differences among methods.

10. In Section 5, Practical Applications, use case of embedding a QR code is interesting. However, a more detailed discussion on limitations (e.g., data volume constraints or real-world scenarios) are missing.

11. In Section 6, Conclusion, Conclude by briefly mentioning future applications in other domains (e.g., video steganography or document watermarking) where MFI-Net could be impactful.

Cite this review as

---

## Round 0.2 · accepted · Accept

Dear authors, we are pleased to verify that you meet the reviewer's valuable feedback to improve your research.

Thank you for considering PeerJ Computer Science and submitting your work.

Reviewer 1 ·

Basic reporting

The authors have addressed all the concerns.
I now accept the work.

Experimental design

The authors have addressed all the concerns.
I now accept the work.

Validity of the findings

The authors have addressed all the concerns.
I now accept the work.

Additional comments

The authors have addressed all the concerns.
I now accept the work.

Cite this review as